# An evaluation protocol of 'Replicability Model' project for detection and treatment of leprosy and related disability in Chhattisgarh, India

**Joydeepa Darlong**[1]*, **Karthikeyan Govindasamy**[1], **Onaedo Ilozumba**[2],
**Sopna Choudhury**[2], **Anjali Shrivastva**[1], **Frances Griffiths**[3], **Samuel Watson**[2], **Jo Sartori**[2],
**Richard Lilford**[2]

1 The Leprosy Mission Trust India, New Delhi, India, 2 Institute of Applied Health Research, College of Medical and Dental Sciences, University of Birmingham, Edgbaston, Birmingham, United Kingdom, 3 Warwick Medical School, A-155, University of Warwick, Coventry, United Kingdom

* joydeepa.darlong@leprosymission.in

## Abstract

**Data Availability Statement:** No datasets were generated or analyzed during the current study. All

### Introduction

People affected by leprosy are at increased risk of impairments and deformities from peripheral nerve damage. This mostly occurs if diagnosis and treatment is delayed and contributes to continued transmission within the community. Champa district of Chhattisgarh state in India is an endemic area with the highest national annual case detection and disability rates for leprosy. The Replicability Model is a system strengthening intervention implemented by the Leprosy Mission Trust India in Champa that aims to promote early diagnosis and treatment of leprosy, improve on-going management of the effects of leprosy and improve welfare for the people affected by leprosy. This protocol presents a plan to describe the overall implementation of the Replicability Model and describe the barriers and facilitators encountered in the process. We will also quantify the effect of the program on one of its key aims-early leprosy diagnosis.

### Methods

The replicability model will be implemented over four years, and the work described in this protocol will be conducted in the same timeframe. We have two Work Packages (WPs). In WP1, we will conduct a process evaluation. This will include three methods i) observations of replicability model implementation teams' monthly meetings ii) key informant interviews (n = 10) and interviews with stakeholders (n = 30) iii) observations of key actors (n = 15). Our purpose is to describe the implementation process and identify barriers and facilitators to successful implementation. WP2 will be a quantitative study to track existing and new cases of leprosy using routinely collected data. If the intervention is successful, we expect to see an increase in cases (with a higher proportion detected at an early clinical stage) followed by a decrease in total cases.

relevant data from this study will be made available upon study completion.

**Funding:** This research was funded by the National Institute for Health Research (NIHR: 200132) using UK aid from the UK Government to support global health research. The views expressed in this publication are those of the author(s) and not necessarily those of the NIHR.

**Competing interests:** I have read the journal's policy and three authors (Joydeepa Darlong, Karthikeyan Govindasamy and Anjali Shrivastva),of this manuscript have the following competing interests: They work for the implementing organization TLMTI

## Conclusion

This study will enable us to improve and disseminate the Replicability Model by identifying factors that promote success. It will also identify its effectiveness in fulfilling one of its aims: reducing the incidence of leprosy by finding and tracking cases at an earlier stage in the disease.

## Background

Leprosy is a neglected tropical disease, caused by a slow growing bacterium called the Mycobacterium leprae [1]. The condition is characterized by anaesthetic skin lesions and damage to peripheral nerves. The clinical presentation varies across a wide spectrum from tuberculoid to lepromatous leprosy. Treatment of leprosy constitutes of a combination of antibiotics (Rifampicin, Dapsone and Clofazimine) called multi drug therapy [2]. If not treated timely and appropriately, it can cause irreversible damage to the peripheral nerves leading to loss of sensation and muscle function, resulting in visible deformities, functional impairments, and disabilities with serious health, social and economic consequences [3].

The number of active cases (quantum of infection) is the major determinant of leprosy transmission in the community. Because of the long incubation period of Mycobacterium leprae, delayed detection results in prolonged contact of infected patients with the other members of thus increasing transmission [4]. Delays in diagnosis and treatment of leprosy occur at three levels: individual, community and health facility. At the individual and community level, a lack of awareness about leprosy and prevailing stigma surrounding the disease results in concealment and delay in reporting to health centres [5]. At the health facility level, a lack of clinical expertise and limited resources lead to further delay and subsequent complications, such as inflammatory reactions and peripheral nerve damage [6].

Since the introduction of multi-drug therapy in the early eighties, there has been a marked reduction in the number of cases of leprosy. The World Health Assembly Resolution in 1991 to "eliminate leprosy as a public health problem by 2000" was followed by a 90% decrease in the prevalence of leprosy, which dropped to less than 1 in 10,000 at the global level [7]. By the end of 2005 most countries had eliminated leprosy at the national level. However, India continues to face challenges with leprosy elimination. In 2020, India registered 114,451 cases. The prevalence rate of 0.4 per 10,000 population represents 57% of the global burden [8]. Of the new cases detected, 58.1% were multibacillary, 5.8% were in children less than 14 years of age, and 2.4% of all cases had visible deformities. There are more than 3 million people with leprosy deformities that need attention and care [9]. A high national case detection rate implies continued transmission of the disease, and a high grade 2 deformity rate is evidence of delayed detection. Thus, finding new leprosy cases early is the accepted cornerstone of leprosy control strategies [10]. The National Leprosy Eradication Program in India is responsible for the diagnosis and treatment of the disease and for prevention and management of disability due to leprosy [11]. National program activities and plans are formulated centrally and implemented by state authorities. The National Leprosy Eradication Program has developed multiple initiatives to promote early detection of leprosy. However, they are rarely evidence-based or integrated into national health programs. Interventions that are evidence-based and implemented through the national health program for leprosy are more likely to be sustainable and replicable elsewhere than stand-alone interventions delivered by an outside agency.

There is therefore a need for a program that comprehensively addresses early diagnosis of leprosy, prevention of leprosy related disability, and socioeconomic advancement and inclusion of people affected by leprosy. Responding to this need, the Leprosy Mission Trust India has implemented a project in the district of Champa, Chhattisgarh state, named the Replicability Model project which seeks to strengthen the activities of the government in the three pillars of zero leprosy: transmission, disability, and discrimination.

In summary, the Replicability Model has two noteworthy features. First it covers leprosy management as a whole covering prevention of spread, management of complications and improving well-being through social integration. Second, it is not a stand-alone NGO delivered program, but it is delivered through the public health service, a feature that should make it replicable.

This manuscript presents the protocol for an evaluation of the Replicability Model project (2020–2024). The research aims to:

i. Find out how the overall Replicability Model project is implemented and identify barriers and facilitators encountered along the way.

ii. Evaluate the first specific aim of the Replicability Model project by measuring effects on early diagnosis of leprosy.

A team from the University of Birmingham and members from The Leprosy Mission Trust India, who are not involved in the implementation of the Replicability Model project, under the NIHR Research and Innovation for Global Health Transformation (RIGHT) grant, will conduct the evaluation.

## Intervention description – The Replicability Model (RM) project

The RM project is co-developed by The Leprosy Mission Trust India (TLMTI) and the ministry of Health and funded by the Leprosy Mission England & Wales. It is implemented by TLMTI in Janjgir-Champa district, Chhattisgarh. This project has an implementation timeline of 5 years (2020–2024), and project evaluation will occur concurrently.

The main objective of the RM project is to strengthen the health system to reduce disability rate amongst people affected by leprosy in Janjgir-Champa District. The RM project has three aims:

1. improve early detection of leprosy

2. improve the prevention and management of leprosy and reduce leprosy disabilities

3. improve access to rights and entitlements and reduce stigma and discrimination to promote inclusion

Our hypothesis is that, due to its comprehensive nature and integration into the existing governance systems, the RM project will prove to be effective, sustainable, and transferable.

The overall framework of the intervention is illustrated in (Fig 1) and described in detail according to the template for reporting interventions (TiDiER) document in S1 File. The project involves several mission critical activities, including health promotion, health worker education, data gathering, health care systems (surveillance, contact tracing, referral) and provision of equipment. The delivery of the RM project involves health workers namely Mitanins (community health workers), male volunteers, Child Health Screening and Early Intervention Services (Rashtriya Bal Swasthya Karyakram) staff and medical officers.

Training sessions will be 'cascaded' to front line staff through a train-the-trainers program as follows: 180 Masters trainers from the study site will attend a 1-day trainers' program. These

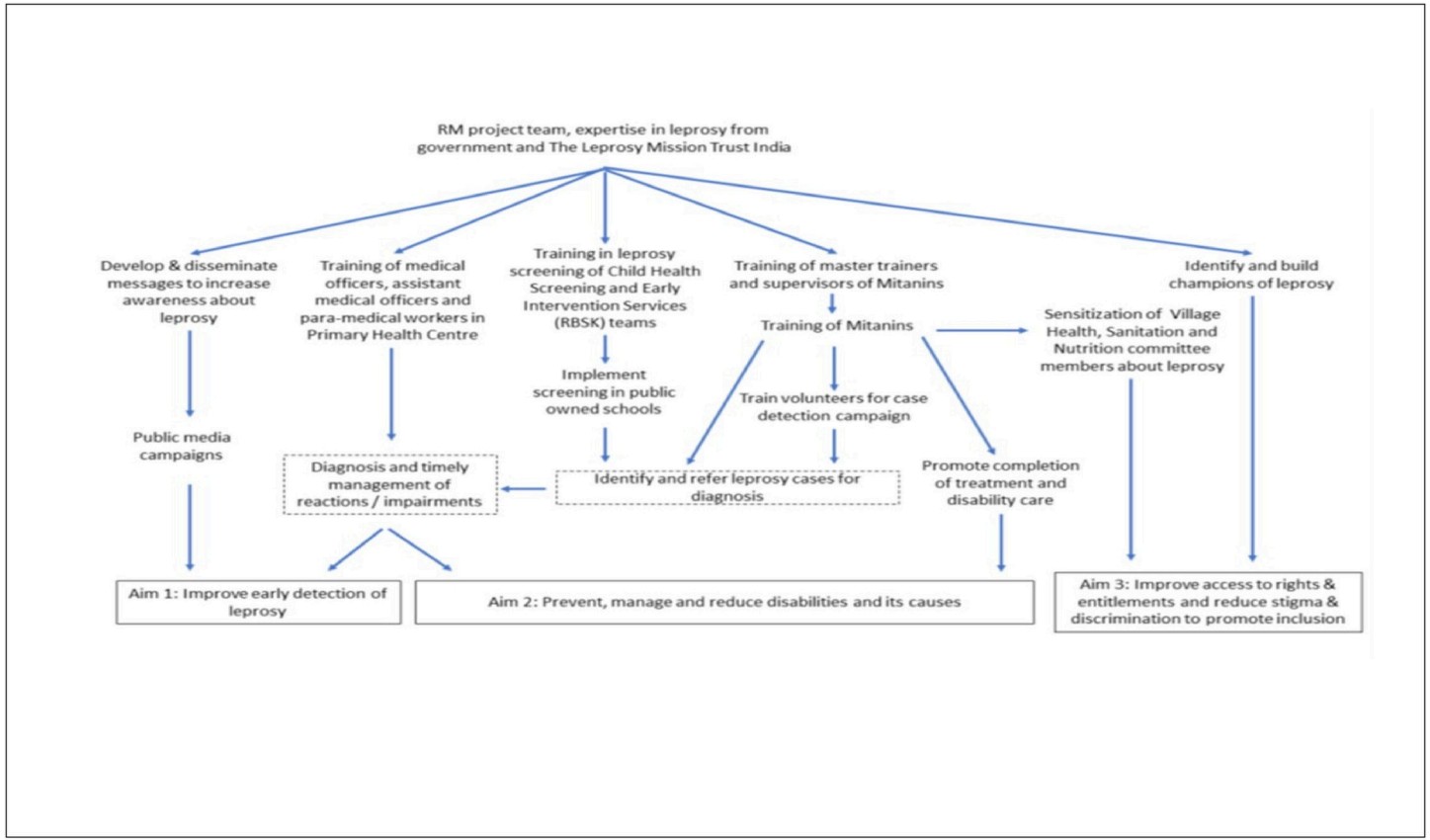

**Fig 1. Overall framework of interventions of Replicability Model in Janjgir-Champa district.**

master trainers will then be responsible for training Mitanins, male volunteers and Rashtriya Bal Swasthya Karyakram team members as follows:

i. Mitanins: Mitanins (n = 3,791) will attend training sessions delivered by the master trainers. The sessions will be delivered to groups of approximately 20 Mitanins and address leprosy detection and the referral process to health facilities for diagnosis and treatment.

ii. Team members and male volunteers: 10 members of the Rashtriya Bal Swasthya Karyakram teams and male volunteers will be trained in a one-off training program in the district head-quarters. If needed, refresher courses will be planned.

iii. Medical officers: The medical officers (n = 7) and assistant medical officers (n = 65) posted in the Primary Health Centres will receive training on diagnosis, confirmation, and management of cases for leprosy and its complications.

## Methods

### Study setting

Chhattisgarh state has 27 districts with a population of 28 million as per census 2011 [12]. With a literacy rate of 70% it is one of the six worst performing states in India. About 45% of the state's population lives below the poverty line, with a major portion (31%) belonging to

Scheduled Caste/Scheduled Tribe category [13]. Tribal populations in India are often disadvantaged and report poor health outcomes, limited access to infrastructure and financial opportunities [14].

We will conduct our study in the district of Janjgir-Champa in Chhattisgarh state. Janjgir-Champa has a population of 1,884,454 as per 2019–2020 report of the national health mission.

The state has the highest burden of leprosy cases and leprosy-related Grade 2 Disability. During the reporting year April 2018 to March 2019, the annual new case rates in Chhattisgarh state (28.3%) and Jangir-Champa districts (38.3%) were higher than the national rate of India (8.7%). During the same period, the proportion of people presenting with a visible disability at the time of diagnosis in Chhattisgarh was higher than the national average (4.4% vs 3.04%), while proportions in Janjgir-Champa district was 3.7%.

In 2019 the control district of Raigarh, adjacent to the study site, had a population of 1,766,678. Like Jangir-Champa district it has a high has a higher number of new cases 1236 vs 724 in Champa. According to the 2019–2020 report of the national health mission the percentage of Grade 2 disability was 1.29% of the population.

The research study will be carried out over the four years of the intervention in two-work packages:

*Work Package 1 (WP1)*: Process Evaluation of the RM project, observing factors that impede and promote its implementation.

*Work Package 2 (WP2)*: Quantitative Evaluation of effectiveness in detecting new cases.

We will conduct a further nested study in which we will study ulcer prevalence and intervene with the intention of reducing the burden of ulcers in the community. This third study nested in the Replicability model and will be described in a separate protocol.

## WP1 – Process evaluation of the RM project

WP1 will evaluate the overall RM project as implemented in Janjgir-Champa, Chhattisgarh. The RM project is complex [15] and, as stated, will be implemented within the existing health system and the communities it serves. The aim is to study how the intervention is implemented and identify barriers and enablers to implementation [16].

**Identification and evaluation of mission critical activities of the RM project.** We define mission critical activities as those that are essential to achieve the project aims. To identify and refine mission critical activities, we will conduct key informant interviews with individuals from stakeholder groups including TLMTI staff, Mitanins, Rashtriya Bal Swasthya Karyakram team members, medical officers, primary healthcare workers and community representatives (n = 10).

The mission critical activities will be evaluated in three ways: observation of team meetings, interviews with key stakeholders and observation of key actors in their workplaces.

1. *Observation of RM Implementation Team Meetings*: The research team will observe monthly meeting where senior staff discuss the implementation of the RM project and make decisions. The researcher will take anonymised field notes of monthly meetings. For these meetings there will be no prescribed observation check list.

2. *Interviews with Key Stakeholders*: The interview schedule will be developed and piloted with representatives from the relevant stakeholder groups (n = 10). Semi-structured interviews will be conducted with 8–10 participants in each of the three groups responsible for intervention implementation as described above and persons affected by leprosy (n = 30). Interview guides will cover the following topics: improvement in knowledge and ability to screen, diagnose, refer, and manage cases thus achieving early detection of leprosy.

3. *Observations of Key Actors*: We will observe 15 Mitanin daily sessions and 5 Rashtriya Bal Swasthya Karyakram team members (n = 20), in their workplace using checklists to evaluate the fidelity of the mission critical activities. The researcher will be trained to observe and record key events including the interactions between health workers and community members, and enactment of the mission-critical activities. Checklists will be used for the observation of key actors.

**Data analysis.** Data analysis of three data types will focus on how the mission-critical activities are (or are not) undertaken by the personnel within their context and factors that facilitate or impede implementation We will examine how different health service or community contexts affect the process. We will use the theoretical approach summarised by May [17, 18].

The key analysis questions will be iteratively refined during process evaluation data collection. The basic framework will include questions such as:

- How, where, when and with whom do the actors enact the mission-critical activities, how does this vary and why?

- What do they negotiate in advance to enable them to enact the mission-critical activities (e.g., changes in working patterns or systems; changing attitudes in communities), how easy is this negotiation and what prevents change?

- How do the mission critical activities become normalised within their daily activities/ context?

- How do relationships within their context change to enable mission critical activities and their sustainability?

## WP2 – Evaluation of RM project aimed at early detection of new cases (aim 1 of RM project)

In this WP a time series analysis will be conducted in the intervention district with a contemporaneous time series in a control district of Raigarh with similar epidemiological trends as the study district. The trends in epidemiological indicators for the last 6 years will form the baseline situation of the disease in the intervention and control areas. Data on all patients registered for treatment is available from the community health centres and recognized non-government organization (NGO) owned hospitals (Fig 2).

## Data

The pre-intervention period will be 2015 – 2020. The data for the post-intervention period will be 2021 and 2024. Individual patient data is collected by the NLEP staff of Community Health Centres routinely using standard individual patient forms (see S1 Appendix) and treatment registers (S2 Appendix). A similar set of data will be collected from an adjoining district which will be the control site, where leprosy burden is similar, but no intervention has been implemented.

The key epidemiological indicators for data analysis are shown in Table 1. Timeline for data collection activities is presented in Fig 3.

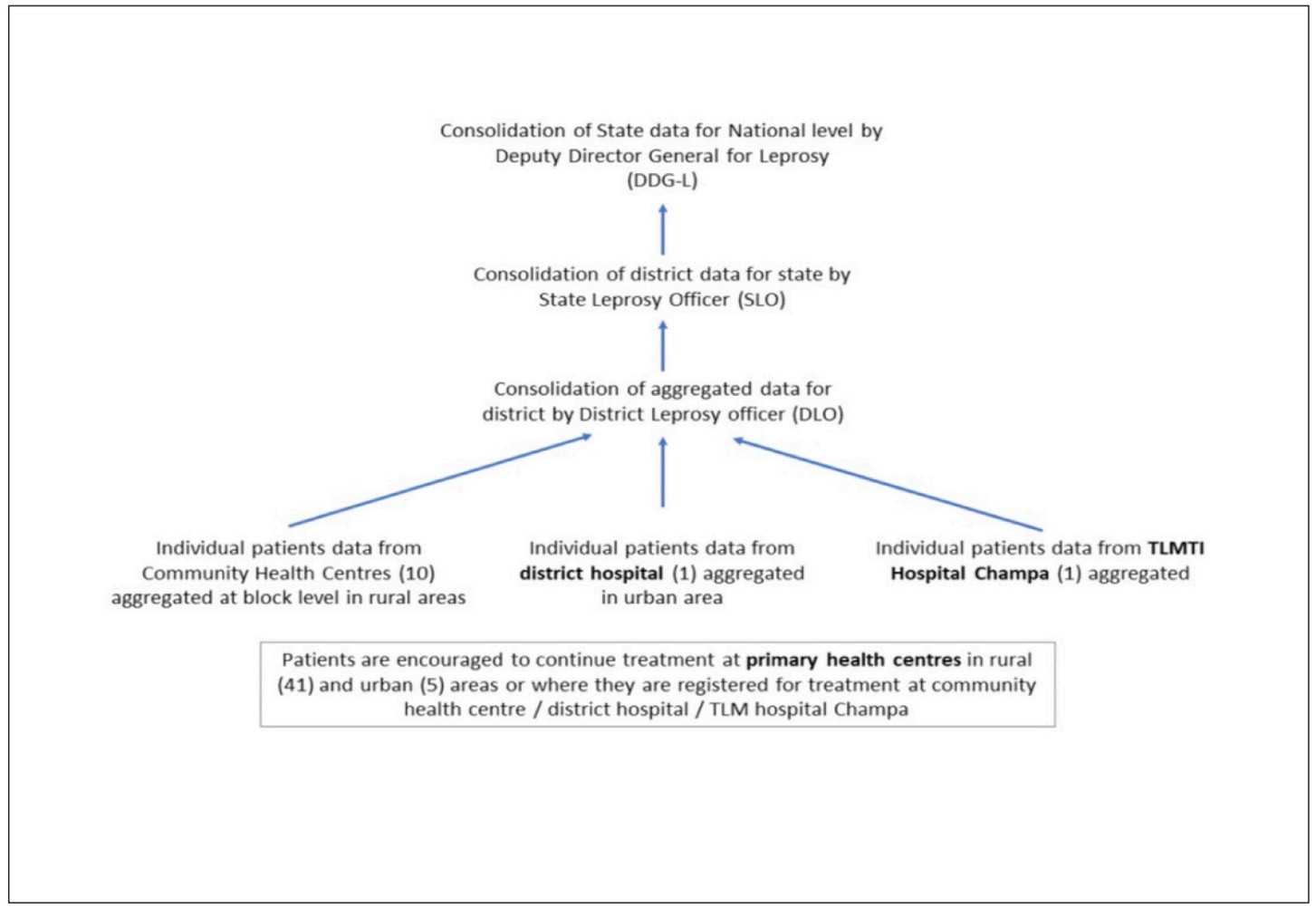

**Fig 2. Shows the flow of data from Primary Health Centres to state level.**

**Table 1. Key epidemiological indicators.**

| Indicators | Numerator | Denominator | Source of data |
|---|---|---|---|
| Prevalence rate (PR) per 10,000 population | Number of cases under treatment at the end of the reporting period | Total population of the area where the data is drawn | Treatment register |
| Annual new case detection rate (ANCDR) per 100,000 population | Number of new leprosy cases detected during the reporting period | | Treatment register |
| Proportion of patients with grade II disability among new cases | Number with grade II disability at diagnosis among new cases detected | Number of new leprosy cases detected during the reporting period | Individual patient card* |
| Proportion of child cases (<15 years) among new cases | Number of child cases (<15 years) among new cases detected | | |
| Proportion of MB cases disability among new cases | Number of MB cases among new cases detected | | |
| Proportion of female cases among new cases | Number of female cases among new cases detected | | |
| Proportion of SC/ST cases among new cases | Number of SC/ST cases among new cases detected | | |

**Fig 3. Timeline for data collection in intervention and control districts.**

### Data entry

This data will be extracted, and anonymous data entered into the project database on a monthly basis by the NIHR RIGHT research project staff. Range limits and logic checks (e.g., for conflicting responses) will be built into the electronic data collection forms to prevent erroneous data entry. Initial data from the first ten participants will be cross-checked by the local lead investigator to ensure that full and accurate data are collected.

### Data analysis

The objectives of the analysis are to assess the impact over time of the intervention on:

i.  the incidence rate of (diagnosed) leprosy and

ii.  the incidence rate of leprosy by grade of disability.

The incidence rate of (diagnosed leprosy): The analysis will be at the block-month level. For block $i = 1, \ldots, N$ at time $t = 1, \ldots, T$ the number of new cases is $y_{jt} \in \mathbb{N}$, which we model as Poisson distributed with intensity $\lambda_{jt}$:

$$y_{jt} \sim Poisson\left(\lambda_{jt}\right)$$

$$\lambda_{jt} = e_{jt}\exp\left(\beta_0 + \beta_1' x_{jt} + \gamma_t I\left(d_{jt} = 1\right) + \alpha_j + \tau_t\right) \qquad (1)$$

where $x_{jt}$ is a vector of block level covariates including demographic indicators like the age distribution of the population, $d_{jt}$ is an indicator for whether block $j$ has started the implementation of the intervention at time $t$, $\alpha_j \sim N\left(0, \sigma_\alpha^2\right)$ is a block-level random effect, and $\tau_t$ is a set of monthly indicators. We include an offset term $e_{jt}$, which is the size of the population in the block so that the exponential term represents a relative risk. We allow the treatment effect $\gamma_t$ to vary by time as we expect the incidence to first increase and then decrease post implementation of the intervention. The roll-out of the intervention will take a number of months so there will be a number of "cross-over" months. We will make comparisons of the parameter of

interest over the whole period and also compare strictly pre- and post-intervention estimates of the parameters as well as intervention time interactions.

The incidence by grade of disability, we use the same model as specified in Equation but set the outcome to be the number of cases at each level of disability. Post intervention we hypothesise that the incidence of cases with severe disability will decrease and those without severe disability to increase in the first instance. Over-time, cases will decrease if the intervention is successful.

## Ethical approval and consent to participate

The research will be performed in accordance with the Declaration of Helsinki for Human Research of the World Medical Association and approval has been granted by the University of Birmingham Biomedical and Scientific Research Ethics Committee (BSREC) (Approval number ERN_20–0816 (reference number linked to multiple related studies) and locally in India by The Leprosy Mission Trust India Ethics committee. (Approval number: C-046/TLMTI EC/21). Any deviations from the approved protocols will be documented.

Eligible people will be provided with a Participant Information Sheet in local languages. Information will be provided verbally for participants who are non-literate. Written informed consent will be obtained from all participants, or thumb/fingerprints will be requested in lieu of a signature if necessary. Translated consent forms will be back-translated according to the WHO recommendations for quality assurance purposes. Participants will be free to withdraw at any time.

## Discussion

The RM Project is an ambitious program targeting leprosy. It is unusual, arguably unique, in its philosophy and scope. First, it is not a stand-alone program. Rather it is embedded in and owned by the health service although largely funded by an NGO. Second, it goes beyond health services, involving schools and aiming to improve case detection among school children and third, it has multiple aims addressing the three zeros of transmission, disability, and discrimination of leprosy.

The RM project as described in this protocol has the potential to influence the incidence rate of leprosy. The first indication of an effective intervention would be an increase in the total number of cases or new cases detected/reported, a downstaging of the disease and subsequently a decrease in a total number of leprosy cases.

This study both capitalises on and contributes to the planned rollout of the RM project. A key strength of this study is its mixed methods approach which will contribute to an understanding not only of outcomes but also processes. Our results will be useful in informing leprosy programs elsewhere and we hope our methods will be useful to inform the evaluation of other multi-purpose, multi-component, across-agency interventions. The evaluation will provide additional evidence relating to the effectiveness of the case-finding approach of the RM project by conducting a comprehensive search for people at risk of leprosy and their accurate diagnosis by the physician in the health centres.

However, there are also study limitations. The evaluation team works for the same organization as the implementation team. While research and management processes will be implemented to ensure that the evaluation is independent, this cannot be completely guaranteed. However, the evaluation team will document all deviations from the protocol, and these will be included in future reporting. In addition, in WP3 (a nested study to be described separately), we will estimate the prevalence of disabilities among people affected by leprosy in the study area. This will provide baseline data to effectively plan, develop, and evaluate a follow-on study

of an intervention to reduce the prevalence of disabilities in eyes, hands, and feet through enhanced self-care in line with aim 2 of the RM project.

## Supporting information

**S1 Appendix. Baseline data collection for early detection of leprosy (WP2).**
(TIF)

**S2 Appendix. Treatment register.**
(TIF)

**S1 File. TIDieR description of replicable model project interventions.**
(DOCX)

**S1 Questionnaire. Inclusivity in global research.**
(DOCX)

## Acknowledgments

We acknowledge the Replicability Model Implementation team for their cooperation in the development of the evaluation protocol. We would also like to acknowledge Sian Arulanantham from The Leprosy Mission England and Wales for her continued support with all our work.

## Author Contributions

**Conceptualization:** Joydeepa Darlong, Karthikeyan Govindasamy, Onaedo Ilozumba, Sopna Choudhury, Frances Griffiths, Jo Sartori, Richard Lilford.

**Funding acquisition:** Samuel Watson, Jo Sartori, Richard Lilford.

**Investigation:** Joydeepa Darlong, Karthikeyan Govindasamy, Samuel Watson, Richard Lilford.

**Methodology:** Karthikeyan Govindasamy, Frances Griffiths, Samuel Watson, Richard Lilford.

**Project administration:** Anjali Shrivastva.

**Supervision:** Joydeepa Darlong, Karthikeyan Govindasamy, Richard Lilford.

**Writing – original draft:** Joydeepa Darlong, Karthikeyan Govindasamy.

**Writing – review & editing:** Joydeepa Darlong, Karthikeyan Govindasamy, Onaedo Ilozumba, Sopna Choudhury, Anjali Shrivastva, Frances Griffiths, Samuel Watson, Richard Lilford.

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
