## [Decision Letter · Decision Letter 0]

7 Dec 2022

PONE-D-22-25961An Evaluation Protocol of ‘Replicability Model’ project for detection and treatment of leprosy and related disability in Chhattisgarh, India.PLOS ONE

Dear Dr. Darlong,

Thank you for submitting your manuscript to PLOS ONE. After careful consideration, we feel that it has merit but does not fully meet PLOS ONE’s publication criteria as it currently stands. Therefore, we invite you to submit a revised version of the manuscript that addresses the points raised during the review process.

Please check reviewer's minor suggestions how to improve your manuscript further - these comments are provided as comments within the pdf file. In addition, please include the ethical approval number granted by the University of Birmingham Biomedical and Scientific Research Ethics Committee (BSREC) as part of ethics declarations.  

We look forward to receiving your revised manuscript.

Kind regards,

János G. Pitter, MD, PhD

Academic Editor

PLOS ONE

Journal Requirements:

5. Please remove your figures from within your manuscript file, leaving only the individual TIFF/EPS image files, uploaded separately. These will be automatically included in the reviewers’ PDF.

6. We note that Figure 2 in your submission contain [map/satellite] images which may be copyrighted. All PLOS content is published under the Creative Commons Attribution License (CC BY 4.0), which means that the manuscript, images, and Supporting Information files will be freely available online, and any third party is permitted to access, download, copy, distribute, and use these materials in any way, even commercially, with proper attribution. For these reasons, we cannot publish previously copyrighted maps or satellite images created using proprietary data, such as Google software (Google Maps, Street View, and Earth). For more information, see our copyright guidelines: http://journals.plos.org/plosone/s/licenses-and-copyright.

Reviewers' comments:

Reviewer's Responses to Questions

**Comments to the Author**

1. Does the manuscript provide a valid rationale for the proposed study, with clearly identified and justified research questions?

Reviewer #1: Yes

2. Is the protocol technically sound and planned in a manner that will lead to a meaningful outcome and allow testing the stated hypotheses?

Reviewer #1: Yes

3. Is the methodology feasible and described in sufficient detail to allow the work to be replicable?

Reviewer #1: Yes

4. Have the authors described where all data underlying the findings will be made available when the study is complete?

Reviewer #1: Yes

5. Is the manuscript presented in an intelligible fashion and written in standard English?

Reviewer #1: Yes

6. Review Comments to the Author

You may also provide optional suggestions and comments to authors that they might find helpful in planning their study.

Reviewer #1: Please see comments in the attachment, needs some revision as in the inserted c omments of the attached manuscript

7. PLOS authors have the option to publish the peer review history of their article (what does this mean?). If published, this will include your full peer review and any attached files.

Reviewer #1: **Yes: **SRINIVAS G

---

## [Author Response · Author response to Decision Letter 0]

14 Apr 2023

1. Report timing, such as annual, midterm, or TERM Evaluation as appropriate may be mentioned Thank you for this comment. We considered the inclusion of timing as suggested. In response we have included modifications in the methods of the abstract (pg. 2, ln 28-29) and (pg. 8, ln 164).

2. How is discrimination component addressed in Replicability model study? Please elaborate.

(Pg 5, ln 86)

Discrimination is documented through Aim 3 of the Replicability model. (pg. 7, Figure 1). This process evaluation is for Aim 1 (transmission)and Aim 2 ( disability) Additional details can also be found in our TiDiER document (supplementary file 1) and intervention description.

3. How is this model innovative for sustainability in compared to other NGO projects delivered through public health/ Govt? (Pg 5, ln 90) We agree with the reviewer that NGO and Public Health collaborations are not new. However, leprosy focused projects tend to be more siloed, either situated within an NGO or the health district. This model is innovative because while the implementation is led by an NGO, all the activities are conducted by district authorities and staff. This means that after the project is completed, there is a scope for the activities continue within the health system. In our experience including our retroactive review of similar projects in India, Nepal and Nigeria, none have followed a similar model.

4. Has Central leprosy division Govt of India been involved in selecting this stakeholder? (Pg 6, ln 98) There was no Government involvement in selecting the University of Birmingham as an evaluation partner. The evaluation has received funding independent of the development and implementation of the Replicability model. The University of Birmingham and partners in three countries including India obtained funding from the National Institute for Health and Care Research (NIHR) Research in the United Kingdom. The funds are directed to projects which aim is to improve self-care in the community for leprosy patients who are at risk of recurrent ulceration and further disfigurement and disability and to better understand the needs of Buruli ulcer patients and the barriers to meeting those needs. The evaluation of the Replicability Model is one project within this larger programme.

5. Timeline (pg. 6, ln 101) Since it’s a process evaluation, the timeline will follow the RM project timeline (2020-2024). This has been included in response to the first comment.

Population of this district, no. of disabled & leprosy disabled in the state & district in last few years. For Comparison, nearby district disability load may be mentioned (pg. 6, ln 109) We have responded to this comment by including the suggested details on pg. 8, ln 151-161

6.Need nearby places (without this intervention) data for comparison (pg. 7, ln 115) The control district for this project Raigarh (pg.8 ln 158 -161 and pg. 11 ln 230) 

7.Need to state the key evaluation objectives (based on intervention objectives/ may be refined) ... The basic framework will include questions such as: How, where, when and with whom do the actors enact the mission critical activities, how does this vary and why? What do they negotiate in advance to enable them to enact the mission critical activities (e.g., changes in working patterns or systems; changing attitudes in communities), how easy is this negotiation and what prevents change? How do the mission critical activities become normalised within their daily activities/context? How do relationships within their context change to enable mission critical activities and their sustainability? The key overall evaluation aims are stated on pg 6 ln 95-99. 

The questions we will answer can be found in WP1(Pg.9 , ln 170 -199) 

The objectives for WP2 are given (pg. 11 ln 216-223)

8.Data sources & plan for Comparison data (pg. 9) The relevant information is provided on (pg. 12 ln 238 -242)

9.Observation check list? (pg. 10, line 126) For these meetings there is no prescribed observation checklist. Rather the observer will take make fieldnotes notes. Observation checklists are used for the observation of key actors (pg. 11, ln 196-213). We have included clarifications in the checklist

10.Timeline for these indicators with comparison area (pg. 13, ln 123) The timeline presented in Figure 4 applies to both the intervention and control groups. The figure caption has been updated to reflect this more clearly

11.Also inform about documentation of deviations from the protocol in full ev report (pg. 16, ln 296) The line has been included on pg. 13, ln 273

12. Approval from State health authorities & Central Leprosy division? (pg. 16, line 297) For this project all permissions were obtained at the State level. The Managing Director of State National Health mission in Champa district invited The Leprosy Mission Trust India to implement this project in the district because of the endemicity of leprosy. We have the documentation of the decision for TLM to implement the project in Champa. 

13.How was it evaluated critically than routinely for intellectual contents? Why do you want to inform that it was critically evaluated? (pg. 17, ln 317) Thank you for this valuable comment about the phrase ‘critically evaluated the intellectual content.’ We considered this phrasing an accurate description of the RJL’s role in developing the protocol and this subsequent manuscript. He is an established leader in Public Health with considerable expertise in methodology of clinical trials, patient safety, service delivery research, and global health. His role in this project went beyond a routine evaluation of the intellectual content. Rather he was fully engaged in ensuring that the protocol and manuscript met the highest academic standards. 

Response to the Editors Comments

Please ensure that your manuscript meets PLOS ONE's style requirements, including those for file naming. The PLOS ONE style templates can be found at The style requirements have reviewed and updated as necessary. 

Please include a complete copy of PLOS’ questionnaire on inclusivity in global research in your revised manuscript. Our policy for research in this area aims to improve transparency in the reporting of research performed outside of researchers’ own country or community. The policy applies to researchers who have travelled to a different country to conduct research, research with Indigenous populations or their lands, and research on cultural artefacts. The questionnaire can also be requested at the journal’s discretion for any other submissions, even if these conditions are not met. We have completed the questionnaire and uploaded as Other. 

We note that the grant information you provided in the ‘Funding Information’ and ‘Financial Disclosure’ sections do not match. The information in Financial Disclosure has been corrected.

Your ethics statement should only appear in the Methods section of your manuscript. If your ethics statement is written in any section besides the Methods, please move it to the Methods section and delete it from any other section. Please ensure that your ethics statement is included in your manuscript, as the ethics statement entered into the online submission form will not be published alongside your manuscript. The ethics statement is now only presented in the Methods section. 

Please remove your figures from within your manuscript file, leaving only the individual TIFF/EPS image files, uploaded separately. These will be automatically included in the reviewers’ PDF Figures have been removed from within the manuscript file

We note that Figure 2 in your submission contain [map/satellite] images which may be copyrighted. All PLOS content is published under the Creative Commons Attribution License (CC BY 4.0), which means that the manuscript, images, and Supporting Information files will be freely available online, and any third party is permitted to access, download, copy, distribute, and use these materials in any way, even commercially, with proper attribution. For these reasons, we cannot publish previously copyrighted maps or satellite images created using proprietary data, such as Google software (Google Maps, Street View, and Earth). For more information, see our copyright guidelines: http://journals.plos.org/plosone/s/licenses-and-copyright. The image has been deleted

Please include captions for your Supporting Information files at the end of your manuscript, and update any in-text citations to match accordingly Captions have been included and in-text citations updated

Please review your reference list to ensure that it is complete and correct. Reference list has been reviewed.

---

## [Decision Letter · Decision Letter 1]

23 May 2023

An Evaluation Protocol of ‘Replicability Model’ project for detection and treatment of leprosy and related disability in Chhattisgarh, India.

PONE-D-22-25961R1

Dear Dr. Darlong,

We’re pleased to inform you that your manuscript has been judged scientifically suitable for publication and will be formally accepted for publication once it meets all outstanding technical requirements.

Kind regards,

János G. Pitter, MD, PhD

Academic Editor

PLOS ONE

Additional Editor Comments (optional):

Reviewers' comments:

Reviewer's Responses to Questions

**Comments to the Author**

1. Does the manuscript provide a valid rationale for the proposed study, with clearly identified and justified research questions?

Reviewer #1: Yes

2. Is the protocol technically sound and planned in a manner that will lead to a meaningful outcome and allow testing the stated hypotheses?

Reviewer #1: Yes

3. Is the methodology feasible and described in sufficient detail to allow the work to be replicable?

Reviewer #1: Yes

4. Have the authors described where all data underlying the findings will be made available when the study is complete?

Reviewer #1: Yes

5. Is the manuscript presented in an intelligible fashion and written in standard English?

Reviewer #1: Yes

6. Review Comments to the Author

You may also provide optional suggestions and comments to authors that they might find helpful in planning their study.

Reviewer #1: The revised draft has incorporated the Suggestions made and the This study will enable tracking cases at an

earlier stage in the disease.

7. PLOS authors have the option to publish the peer review history of their article (what does this mean?). If published, this will include your full peer review and any attached files.

Reviewer #1: **Yes: **SRIINIVAS GOVINDARAJULU

---

## [Editor Report · Acceptance letter]

29 May 2023

PONE-D-22-25961R1 

An Evaluation Protocol of ‘Replicability Model’ project for detection and treatment of leprosy and related disability in Chhattisgarh, India. 

Dear Dr. Darlong:

I'm pleased to inform you that your manuscript has been deemed suitable for publication in PLOS ONE. Congratulations! Your manuscript is now with our production department. 

Kind regards, 

on behalf of

Dr. János G. Pitter 

Academic Editor

PLOS ONE